# Mechanical Harvesting of Ornamental Citrus Trees in Valencia, Spain

**Antonio Torregrosa [1,*], José María Molina [1], Montano Pérez [1], Enrique Ortí [1], Pilar Xamani [2] and Coral Ortiz [1]**

[1] Department of Rural Engineering, Universitat Politècnica de València, 46021 Valencia, Spain; jomohi@upvnet.upv.es (J.M.M.); montano@dmta.upv.es (M.P.); eorti@dmta.upv.es (E.O.); cortiz@dmta.upv.es (C.O.)

[2] Department of Agricultural Ecosystems, Universitat Politècnica de València, 46021 Valencia, Spain; pixamon@gmail.com

[*] Correspondence: torregro@dmta.upv.es

**Abstract:** Citrus trees are used as ornamental plants in several Spanish cities. They give a nice color to the streets and a nice scent in the flowering stage, but when the fruits fall, they dirty the roads and pavements, and can cause accidents; this is the reason why gardeners must detach and collect the fruits. This task is being done manually, but it is quite inefficient and expensive. In this study, three types of machines have been used to mechanize this task: a trunk shaker with umbrella, a trunk shaker hitched to an orchard tractor, and an experimental smaller sized shaker that can be attached to small and pedestrian tractors. The shaking patterns used by each equipment, detachment percentages, mobility constraints, and tree damage have been measured, and reduction costs have been estimated. After three years of study, the system has been fully accepted by both gardeners and citizens.

**Keywords:** vibration; shaker; '*Citrus × aurantium*'

## 1. Introduction

Citrus are used as ornamental trees in several Mediterranean cities because they are well adapted to the ecological conditions, they permeate the atmosphere with a pleasant smell of orange blossom, and in winter, the orange fruits give a beautiful color to the parks and promenades. However, when the fruits start to fall on reaching maturity, they dirty the streets, cause problems for pedestrians, and cause accidents for cyclists, and this is why gardeners have to remove the fruits when the falling process starts [1].

In the city of Valencia, there are over 15,000 trees from the specie *Citrus × aurantium* L. registered (based on personal communication from gardening technicians of Valencia town council), and this high number incurs significant economic cost because at the moment the accompanying tasks are performed manually.

Mechanical detachment by vibration of citrus trees is a proven technique in agricultural citrus [2–5], used for both industry and fresh market, noting that detachment percentages of around 80% are normal with this technique.

The most common techniques used are canopy shakers, that beat all the vegetation with poles, and with this method it is possible to detach the majority of the fruits. There are some experimental low size machines [5], but the present commercial machines are too big in size to be used in urban streets [6]. Hand-held branch shakers are other kind of machines that can help to increase the work capacity, but they are not ergonomic and the workers do not accept them [7]. Therefore, trunk shakers seem to be the most adequate equipment, at present, to detach the fruits of these special trees.

The most suitable vibrational parameters have been studied, and the possible damages to the trees bark, leaves, and trunks have also been analyzed, finding that if the duration of the vibration is not excessive, there are not, usually, any problems observed on the trees. Low frequencies (5–10 Hz) combined with high amplitudes, more than 0.1 m, are highly effective in fruit detaching with low defoliation. However, this can only be applied to trees with long trunks, attaching the shaker at a high point. When the shaker must be applied to the trunk closer to the ground, the amplitudes must be reduced to between 20–30 mm. As a consequence, to obtain a good detachment percentage, the frequency must be increased (15–25 Hz), but the shakes must be of short duration, of less than 10 s to avoid excessive defoliation [7–11].

In order to increase the detachment percentages, chemical abscissors have been tested [12–16], but the small increases in detachment that are reached do not compensate for negative effects such as defoliation, nor consider the increasing social rejection of chemical agents in agriculture and in urban areas.

Despite the vast experience accumulated in the mechanical harvesting of citrus trees, there is no knowledge of the impact of these techniques on special kinds of trees like the urban citrus trees, because they are cultivated in different soil conditions to the agricultural ones, and their architecture is different, with slim trunks, which are nearly 2 m tall.

Thus, the main objective of this study was to explore the feasibility of mechanical harvesting in ornamental citrus trees, reducing harvesting costs and without damaging the trees.

## 2. Materials and Methods

Four harvesting systems were tested: (1) manual; (2) trunk shaker with manual picking of fallen fruits; (3) trunk shaker with umbrella; and (4) experimental light shaker (SMTA) mounted on a pedestrian tractor and with oil supply provided by the external sources of two tractors.

### 2.1. Harvesting Systems

#### 2.1.1. Manual

Workers used poles with a hook at the end to detach the fruit, and later they collected the fruits by hand with brooms and buckets before finally unloaded the buckets into a lorry or trailer (Figure 1).

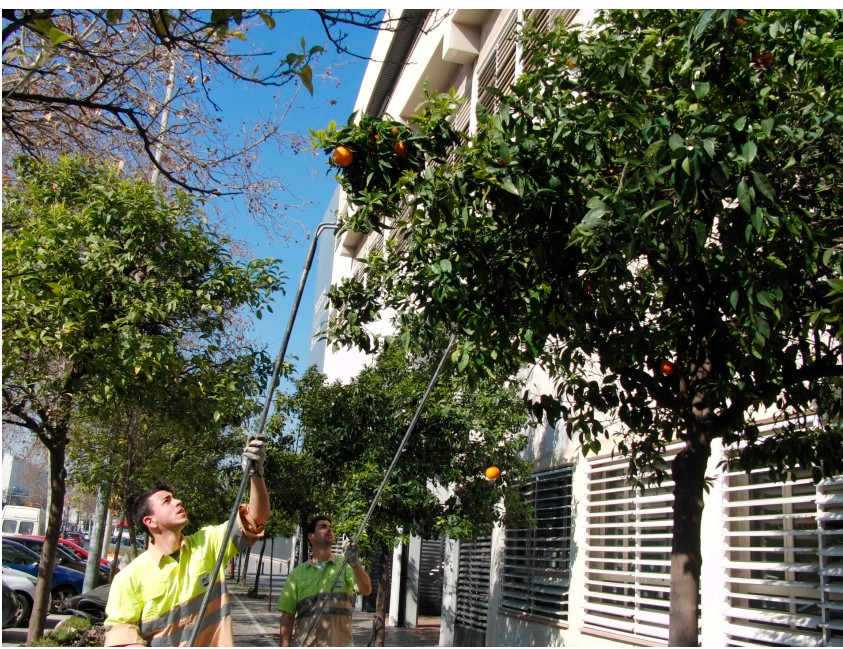

**Figure 1.** Equipment used in manual harvesting.

### 2.1.2. Trunk Shaker

An orchard tractor mounted trunk shaker (Topavi, model 'light shaker', Maquinaria Agrícola Garrido s.l. (Topavi), Autol, Rioja, Spain. www.topavi.es) was used. It was arranged in two parts, one consisting of the oil tank and pumps, attached at the tractor's rear 3 point hitch (with a mass of 640 kg), and the other part (with a mass of 730 kg) was coupled to the front 3 point hitch, and included an extendable arm and clamp with two moving fingers, as well as the hydraulic motor that drives an eccentric mass of 16 kg, and an eccentric radius of 0.13 m that produces an orbital vibration. The tractor was a 66 kW, four-wheel drive orchard tractor Lamborghini Plus 990 F (Figure 2).

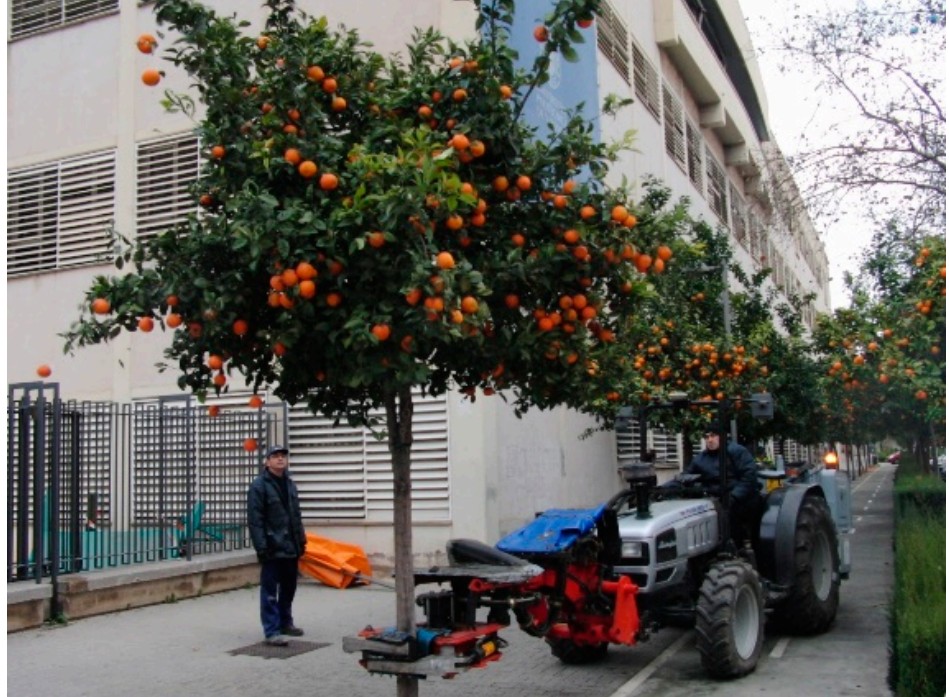

**Figure 2.** Trunk shaker.

The shaker held the tree trunk at 0.8 m height and one to three shakes were applied, depending on to the amount of fruit remaining on the tree after each shake. The duration of each shake was 3 s.

### 2.1.3. Trunk Shaker with Umbrella

This equipment was an almond and olive shaker with umbrella from the trademark 'Estupiña' model PHK-8 (Estupiña, Alcañiz, Teruel, Spain; www.estupi~na.com). It was attached to the tractor in two parts, at the rear of the tractor the tank and pumps of the hydraulic system are hitched and connected to the tractor power take off; at the front, the arms of a 'Tenias' model 'T10 Evolution' loader serve to hitch the umbrella and the shaker (Tenías, Ejea de los Caballeros, Zaragoza, Spain; www.tenias.com).

In 2016 the umbrella was equipped with 3 m long radial poles, obtaining an umbrella of excessive size to operate in most of the street trees, because of this, in the following seasons, a new umbrella was constructed with 1 m long poles, that was more manoeuvrable, despite not catching all the fruits in some cases (Figures 3 and 4). The equipment was mounted on a standard four wheel drive tractor 'Kubota' model 'M110GX-II' 88 kW (Kubota, Osaka, Japan; www.kubota.com). The dimensions of the tractor-harvester set with the folded umbrella were 2.4 m wide and 9 m long.

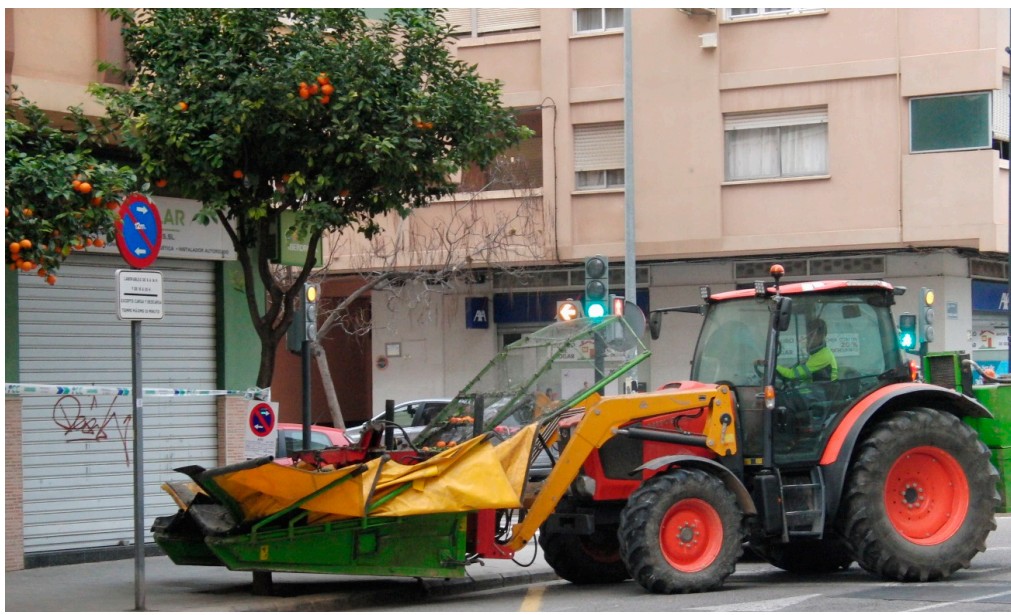

**Figure 3.** Overall view of Estupiña shaker with the small size umbrella.

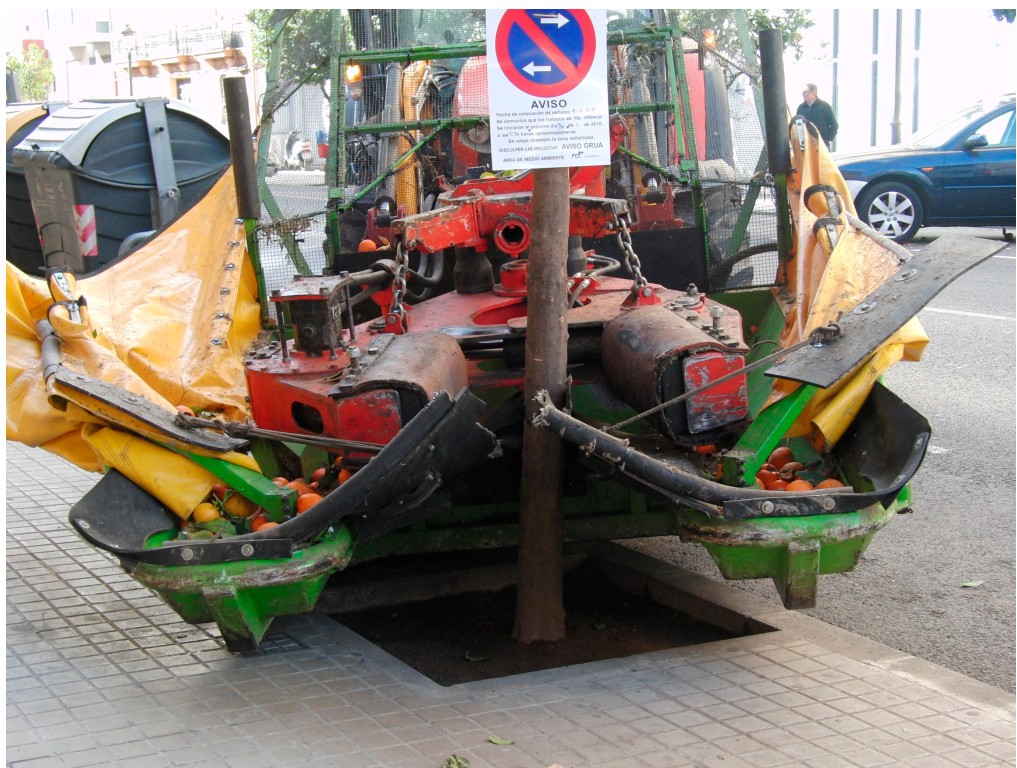

**Figure 4.** Estupiña shaker, close up view of the shaker.

The size of this equipment is its main disadvantage (6.7 m long × 2.4 m wide with the umbrella folded, and 7.7 × 4 m with the 2 m radius when the umbrella is unfolded); it must travel almost exclusively along the street, being very aware of parked cars, and urban furniture close to the trees, buildings and other constructions. Moreover, the trunk shaker must clamp the trunk close to the ground (0.7 m) to have enough clearance for the shaker and the umbrella.

As an advantage, it is able to unload the oranges directly onto a lorry, and can also be used as an intermediate loader to elevate the oranges that are handpicked and transported in 'buckets' by the auxiliary operators.

### 2.1.4. Light Shaker (SMTA) Coupled to a Pedestrian Tractor

This was a light weight, linear and low cost, experimental shaker, that has been already tested in a previous study [17], but that received some improvements to the clamp to avoid damages to the tree bark. In this case, the clamp was made of two steel fingers covered with 60 mm thick rubber pads. The fingers were moved by a hydraulic cylinder.

The shaker was hitched to the forks of a pedestrian hydraulic tractor (Hinowa, Nogara, VR, Italy; Hinowa.com) 'Hinowa' model 'HS 1100' provided with a fork elevator (Figures 5 and 6). The shaker was powered by a hydraulic motor, that received the oil from the external supplies of tractor Lamborghini Plus 990-F that gave a flow of 21 L min$^{-1}$ at 100 bar, or from a John Deere 5820 tractor that gave a flow of 26 L min$^{-1}$ at 100 bar.

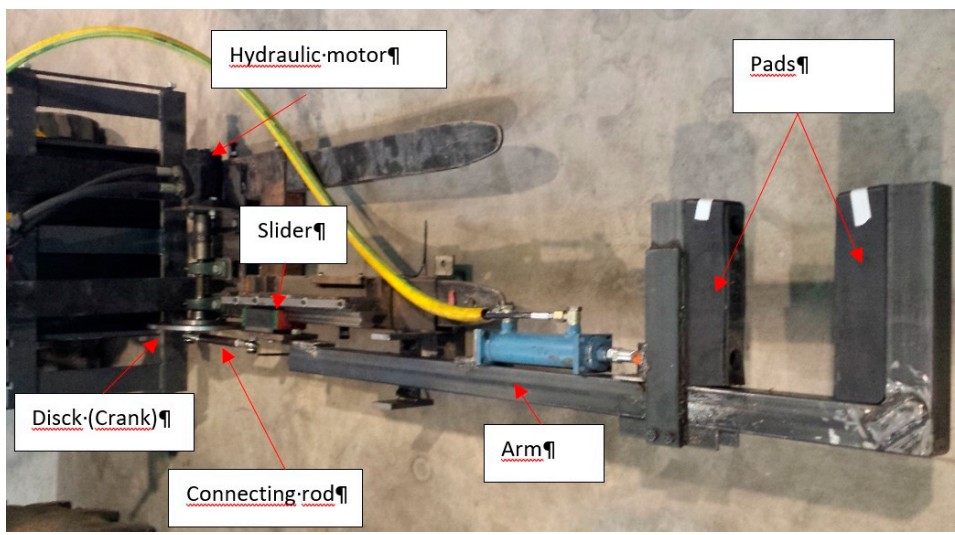

**Figure 5.** Experimental light shaker (SMTA) shaker elements.

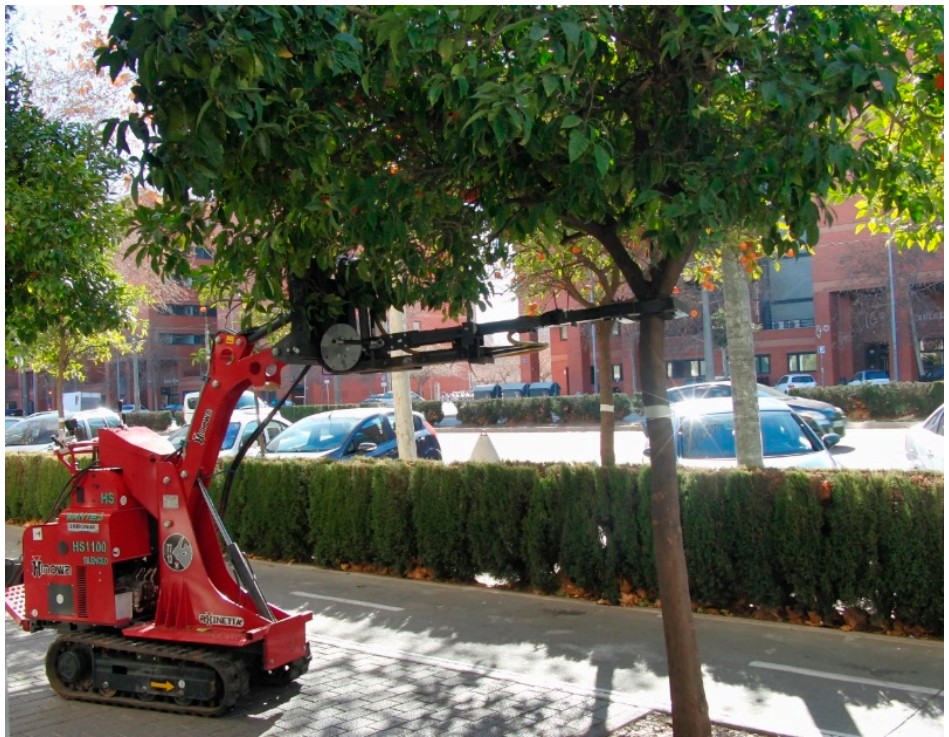

**Figure 6.** SMTA shaking a tree.

## 2.2. Trees and Fruits

Ornamental citrus trees planted in Valencia are mainly from the specie '*Citrus × aurantium*', and usually have a long trunk, nearly 2 m tall, to allow passing pedestrians and vehicles under the crown of the tree. According to the kind of traffic on the street, two typical tree sizes can be found; a) in main avenues, with bus lanes, where the dimensions are 3.0 m from the ground to the end of the trunk, 6.1 m to the tree top, with a 3.8 m crown diameter, and 15 cm trunk diameter at 1.5 m height; and b) in streets with light traffic (cars or pedestrians), where the dimensions are 2.1 m cross height, with a 4.8 m crown top, 2.5 to 3.0 m crown diameter, and 11 cm trunk diameter at 1.5 m height (Figure 7).

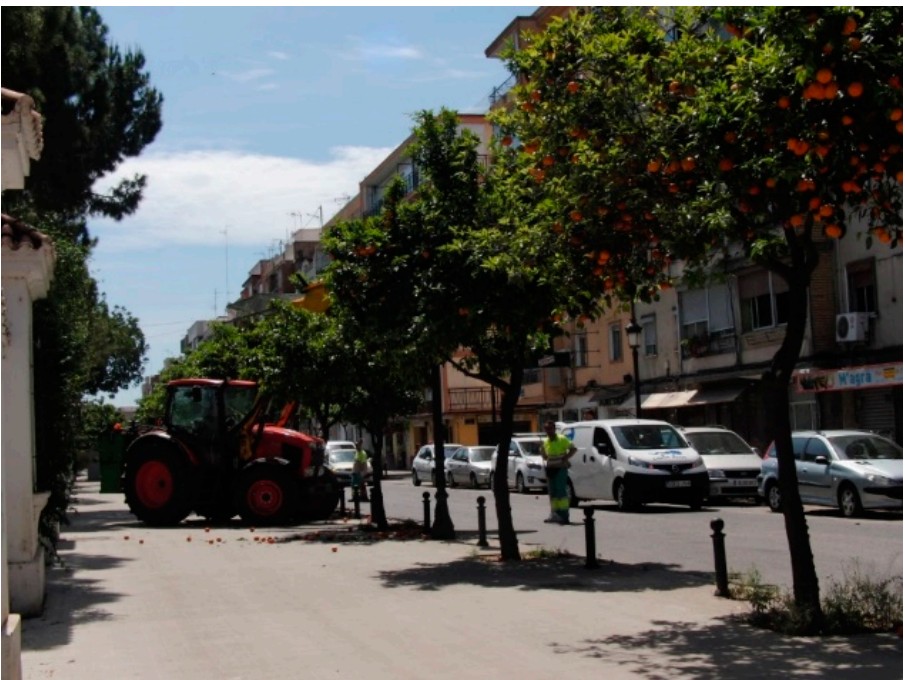

**Figure 7.** Orange trees in Av. Burjassot of Valencia.

Between January and the beginning of May, at each harvesting test, 15 fruits were detached with a dynamometer (500 N maximum force and 0.2 N accuracy) to measure the fruits traction force, and subsequently weighed in the laboratory. Fruit mass ranged from 165 g (sd ± 35 g) and traction force oscillated between 38 and 50 N.

## 2.3. Trees Placing in Valencia (Spain)

Tested trees were placed in:

1. The trees harvested with trunk shaker and SMTA were placed in Av. Tarongers (Googlemaps coordinates 39.479410, -0.342584).

2. The trees harvested with shaker plus umbrella were placed in several streets and avenues in the North sector of the city of Valencia (between the Turia river and the North border of the city). The zones were the field capacity was measured were:

   - Zone: Av. Emilio Baró. This zone is a long avenue (1100 m) with cars parked on one-side and a bus lane on the other. The advantage of working on the trees near the bus lane is that there are no parked vehicles and the machine can move freely. On the other side, some drivers do not move their cars, which makes the harvesting of the trees difficult.
   - Zone: Benimaclet. This area is characterized by short streets (100–300 m), with vehicles parked on one or both sides of the street, some of which were not removed prior to the passing of the machine, thereby causing more interruptions to the work.

- Zone: Av. Naranjos. This zone is a line of trees placed parallel to the tram lane, there are no problems with parked cars, but as the work is done at the same time as trams passes, operators must pay attention to the tram and the umbrella can not be opened, meaning that most of the oranges fall directly to the ground.

The number of trees used in each test are summarized in Table 1.

**Table 1.** Number of trees used to measure equipment working capacity, and to measure the applied vibrations.

|  | Working Capacity | Detachment Percentage | Vibration |
|---|---|---|---|
| Equipment | Number of Tested Trees (year) | Number of Tested Trees | Number of Tested Trees |
| Manual | 10 | — | — |
| Umbrella shaker | 69 (2017–2018) | 10 | 3 |
| Trunk shaker | 13 (2016–2017) | 13 | 5 |
| SMTA | 13 (2017–2018) | 13 | 13 |

## 2.4. Measuring Instruments

The following materials were used to register and measure the results of the trials:

- Action camera 'Go-Pro' model 'Hero-4' (GpPro, Inc., San Mateo, CA, USA; www.gopro.com) to record all the movements of the machines in order to measure the time necessary to accomplish each operation.
- Camera 'Casio' model 'Exilim Pro EX-F1' (Casio Computer Co., LTD, Tokyo, Japan; www.casio.com) to take pictures of the trees before and after shaking to make estimations of detachment percentages. It was also able to record tree movement at 300 fps and so provided a better characterization of the vibration process and to measure the detachment rhythm.
- A triaxial accelerometer and recorder (Gulf Coast Data Concepts, LLC, Waveland, MS, USA; http://www.gcdataconcepts.com), 'US Coast GCDC X200-4', that recorded the vibrations at 400 Hz, was attached to the trunks close to the shaker clamps.
- A dynamometer 'Andilog Center model CNRxx250 (Andilog Technologies, Chaville, France)' to measure the fruits' traction forces.

## 2.5. Detachment Efficiency Estimation

Detachment percentages obtained with the SMTA and Topavi equipment were measured by collecting and counting all the fruits detached, as well as the fruits remaining after shaking.

The detachment percentage was calculated as:

$$D = 100 \, F \, T^{-1} \tag{1}$$

where

$D$, was the detachment percentage (%),
$F$, the number of detached fruits,
$T$, the total number of tree fruits.

## 3. Results

### 3.1. Accelerations

#### 3.1.1. Trunk Shaker

Several short shakes (usually three), of 3 s duration each were applied, clamping the shaker at 0.9 m above the ground. The vibrations had a variable frequency of between 10–21 Hz and 22–26 mm displacement pick to pick, measured on the trunk at 1.8 m height (Figure 8). This shaker produces an orbital vibration with variable frequency. Fruit fell almost vertically, which is interesting because the fruits are not projected towards the buildings and cars near the trees.

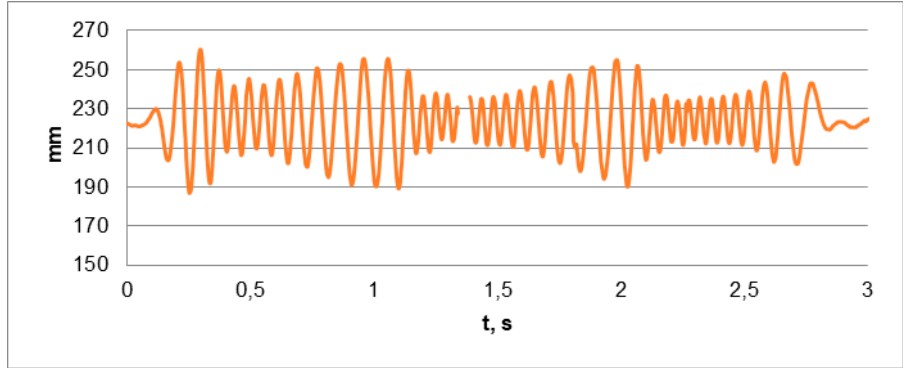

**Figure 8.** Trunk displacement at 1.8 m above the ground in a typical vibration of a Topavi trunk shaker.

#### 3.1.2. Trunk Shaker with Umbrella

On the first day the equipment started working at too high a revolution because the driver was used to working with olive trees, where high frequencies are usual. The frequency in the stable zone was 32 Hz and the trunk displacement, measured on a trunk 237 mm diameter, at 0.5 m above the ground was 13 mm pick to pick. This vibration of the high frequency and low displacement is not the most effective for citrus trees, as in these crops it is more efficient to use lower frequencies and higher displacements [9,10].

After this initial trial, the driver reduced the engine speed, and worked at 11–12 Hz. Additionally, instead of using a long continuous shake, he applied three to five short shakes, of 2–3 s each (Figure 9), in order to serve the transient higher trunk displacements that happen at the start and end of each shake. Using this method, the displacements measured 43 mm compared to the 18 mm in the stable zone (measured at 1.9 m above the ground), observing that this shaking method was more effective (Figure 10).

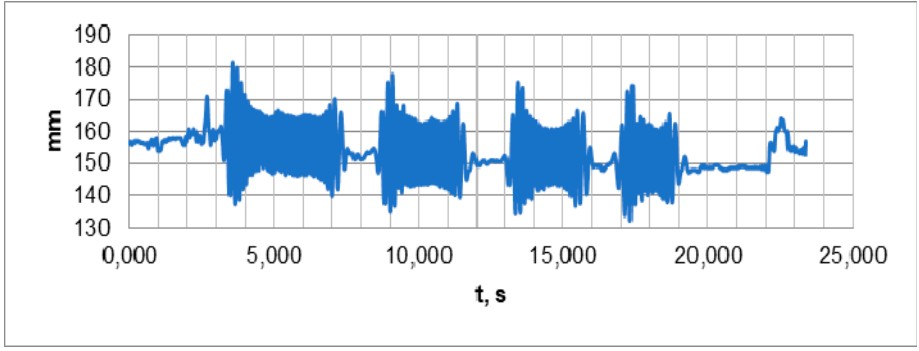

**Figure 9.** Displacement of a trunk, when applying four short vibrations set with umbrella shaker.

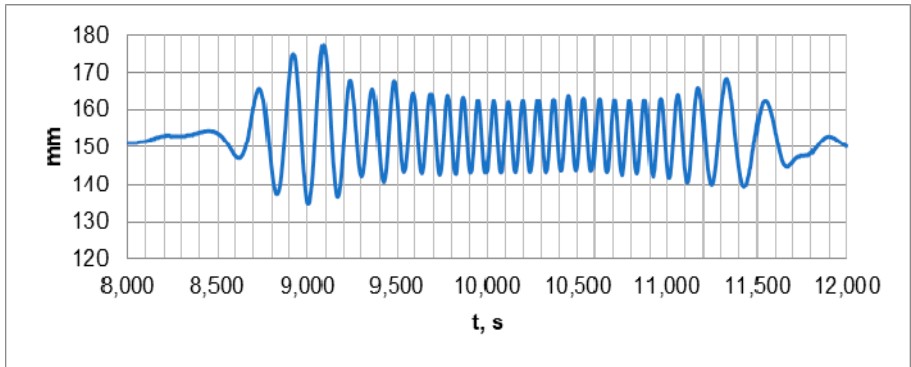

**Figure 10.** Detail of the displacement of the trunk during the second shake of the above figure. Notice how the displacements in the start are double than those in the stable zone.

### 3.1.3. SMTA/ Lamborghini Plus 990-F

The shaker was hitched to the forks of a pedestrian loader, but the motion of the shaker's hydraulic motor was carried out by the external hydraulic supply of the tractor Lamborghini Plus 990-F, that was set to 2500 rpm, obtaining a vibration of 4.0–4.5 Hz. Trees were shaken for 31 s in total, each vibration being a set of five short shakes.

### 3.1.4. SMTA/John Deere 5820

The vibration was also measured in other, similar, trees but using the oil from the external supply of a John Deere 5820 tractor that has a higher flow. In this case, with the tractor engine at 1000 rpm, the shaker reached a frequency of 6 Hz. Accelerations measured on the shaker arm were roughly 120 ms$^{-2}$ peak to peak (Figure 11), and the trunk displacements were at 1.9 m above the ground reaching 293 mm peak to peak.

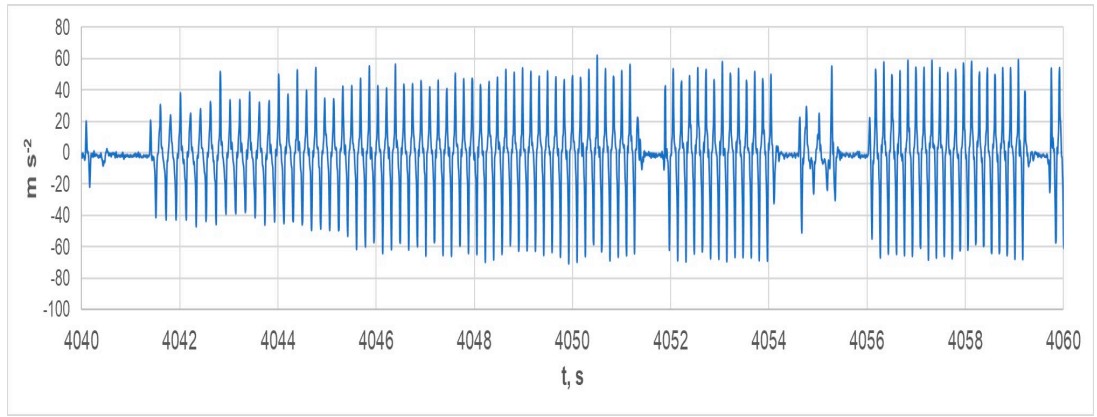

**Figure 11.** Accelerations on the SMTA shaker arm measured at 1.9 m above the ground when powering the shaker with the John Deere tractor external oil supply.

### 3.1.5. Comparison of the Vibrations Reached with Each Equipment

The trunk with umbrella shaker (Estupiña) was the equipment that clamped the trunk closest to the ground, because the umbrella prevents doing it at a higher point, and the trunk displacements measured at 1.8 m above the ground ranged between 10 and 17 mm. The trunk shaker (Topavi) clamped the trunk at 0.9 m, and the displacements measured at 1.8 m above the ground were substantially higher (38–66 mm), and finally, the SMTA shaker was the equipment that clamped the trunks at the highest point (1.9 m) and reached the highest amplitudes, 125–170 mm; this high displacement can only be achieved, without the risk of breaking the trunk, when the clamping point is far from the

ground. Moreover, this elevated clamping point allows the use of low power shakers, as demonstrated by [18]. In Table 2 the main parameters are summarized.

**Table 2.** Main parameters of the vibrations.

| Equipment | Trunk Diameter mm | Clamping Point Height, m | Trunk Displacement at 1.8 m above the Ground Peak-Peak, mm | | Hz | Acceleration Peak-Peak, m s$^{-2}$ | |
| | | | Max | Stable Zone | | Max | Stable Zone |
|---|---|---|---|---|---|---|---|
| Umbrella shaker | 160 | 0.6 | 17 | 10 | 12 | | |
| Trunk shaker | 120 | 0.9 | 66 | 38 | 10–21 | | |
| SMTA/Lamborghini | 160 | 1.7 | 150 | 125 | 4–5 | 120a | 100a |
| SMTA/John Deere | 150 | 1.8 | 170 | 140 | 7 | 325a | 270a |

[a] accelerations measured at the shaker clamp.

## 3.2. Detachment and Collecting

### 3.2.1. Trunk Shaker

This equipment detached more than 80 % of the fruits (81%, sd ± 8%) in a sample of 13 trees.

### 3.2.2. Trunk Shaker with Umbrella

This equipment is the one currently used by the company that collects the fruit from ornamental citrus trees in the North of Valencia. The equipment was used during the years 2017, 2018, and 2019.

In May 2017, the detachment percentage of 13 trees was measured, obtaining an average value of 2% (sd ± 7%). Usually seven or eight operators work simultaneously, one tractor driver and a team to control the traffic, take the fallen fruits out of the umbrella, and even go over fruits that haven't fallen. The shaker tank is also used as a recipient to empty the workers' buckets into, and later the tractor unloads the fruit into a lorry.

### 3.2.3. SMTA

The detachment percentage was measured in the same trees that were harvested with the trunk shaker. The percentages reached were 81% (sd ± 9%) in 2018, and 85% (sd ± 10%) in 2019.

Additionally, the detachment of 10 trees was recorded with camcorder at 300 fps to measure the detachment rhythm. It was noticed that after 15 s, no more fruits fell from some trees, and after 25 s fruit detachment was negligible (Figure 12). The detachment rhythm was slower than that observed with other equipment that uses higher frequencies [9]. The accumulated detachment percentage (*%accumulated*) and the shaking time (*t*) were related, obtaining the following Equation (2), with a determination coefficient $R^2 = 97\%$:

$$\%accumulated = 100 \sqrt{t} \qquad (2)$$

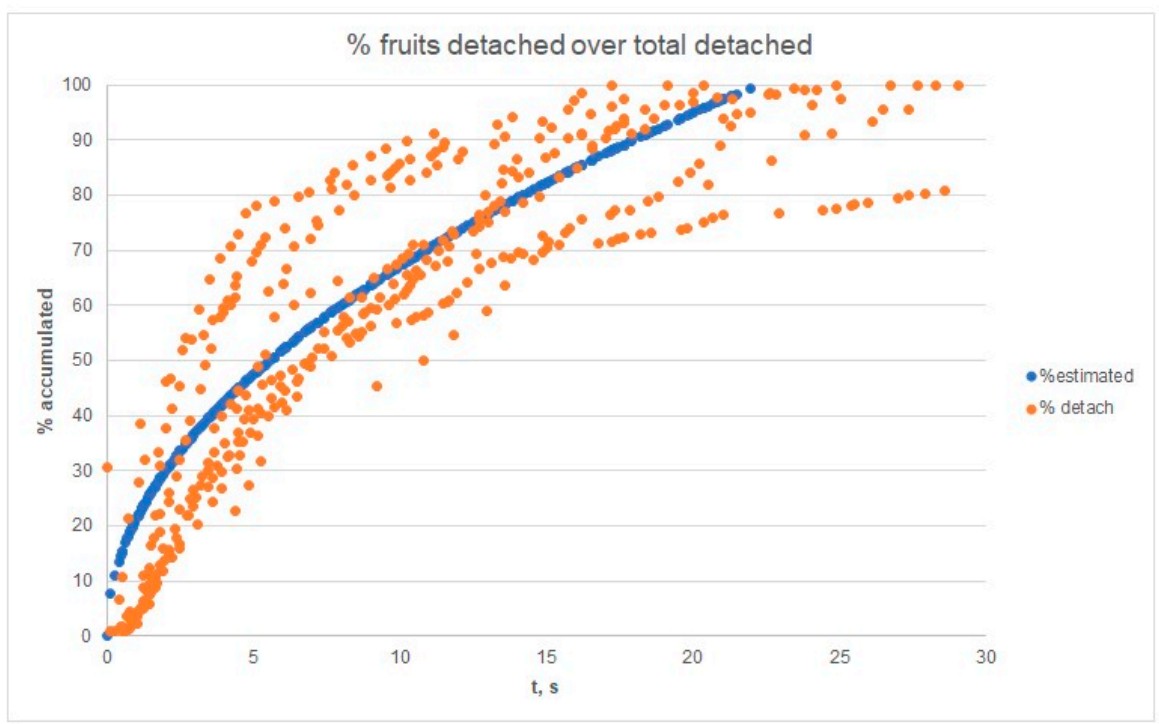

**Figure 12.** Detaching rhythm of 10 trees shaken with SMTA at 4.4 Hz.

*3.3. Field Capacity*

3.3.1. Manual

The working process was analysed from video records, and it was seen that the detaching process was done at a rate of 5 fruit min$^{-1}$. This task can be improved if the operators leave the most difficult fruits to detach on the tree, but they usually try to detach all the fruits.

Picking up the fruits from the ground and putting them into a bucket was done at a rate of 23 fruit min$^{-1}$.

The trees had a highly variable number of fruits, but on average, the trees harvested with SMTA had 500 fruit tree$^{-1}$, considering that the 80% of fruits were harvested, that means 400 fruit tree$^{-1}$, so:

Detaching rate (*DR*) was:

$$DR = 400 \text{ fruit tree}^{-1} \times 0.2 \text{ min fruit}^{-1} = 80 \text{ min tree}^{-1} \equiv 0.75 \text{ trees h}^{-1} \tag{3}$$

Picking rate (*PR*) was:

$$PR = 400 \text{ fruit tree}^{-1} \times 0.04 \text{ min fruit}^{-1} = 16 \text{ min tree}^{-1} \equiv 3.75 \text{ trees h}^{-1} \tag{4}$$

3.3.2. Trunk Shaker

The harvesting process with this equipment consists of two independent operations: (a) fruit detaching, and (b) fruit collecting and transferring to the transport unit.

In the first operation 11 trees were shaken in 13 min, so, the productivity was 51 trees h$^{-1}$, although almost two additional workers are needed to control traffic and pedestrians.

In the second operation, a variable number of workers are needed, depending on several factors such as the amount of detached fruit, distance to the lorry, and other external factors. Although this operation is independent from the detaching, it is convenient to work simultaneously with the first in order to keep the street clean after the harvesting team passes. In our experience, a team of 14 collection

workers will be necessary to work at the same rate as the shaker, consequently the productivity of this system will be:

(1)    tractor with shaker: 51 tree h$^{-1}$
(2)    workers for traffic control + 12 workers collecting and transporting the fruit to the lorry: 51 tree h$^{-1}$, and the equivalent for one operator:

$$51 \times 14^{-1} = 3.64 \text{ tree h}^{-1}. \tag{5}$$

### 3.3.3. Trunk Shaker with Umbrella

This equipment has already been used for three years in Valencia. Work capacity was measured in several zones of the city. A team of seven operators usually works simultaneously with the shaker to control the traffic, pick up fruits that have fallen out of the umbrella, and other assignments.

- Zone: Av. Emilio Baró

The time necessary to harvest 16 trees ranged between 1 and 2 min/tree$^{-1}$. Machine unloading was done every eight trees and took 5 min. As the tractor moved along the street, some interruptions happened which took 2 to 7 min each. Considering all the times, 16 trees were harvested in 40 min, which is 2.5 min tree$^{-1}$.

- Zone Benimaclet

In this case 38 trees were harvested in 137 min, meaning 3.6 min tree$^{-1}$.

The harvesting process of 38 trees in Benimaclet was analyzed in parts. The full process of approaching the tree, trunk clamping, umbrella opening, shaking, umbrella closing, and moving to the next tree took 1.6 min tree$^{-1}$ (sd ± 0.4 min tree$^{-1}$), meaning 43% of total time. The unloading operation took 0.53 min tree$^{-1}$ (sd ± 0.60 min tree$^{-1}$), which is 15% of total time. However, the interruptions of the work for several reasons waste 42% of the total time, with a high variability (1.5 min tree$^{-1}$ on average and sd ±2.6 min tree$^{-1}$).

- Zone Av. Naranjos

Work capacity was 3.0 min tree$^{-1}$ on average and sd ± 1.8 min tree$^{-1}$. So, the working capacity of the shaker with umbrella, in the case of well-organized work that minimizes lost time, was 2.1 min tree$^{-1}$, but in the unpredictable and complicated scenario of city streets the time lost can equal time used productively. In fact, Benimaclet was the zone where this equipment reached the lowest productivity, with 3.6 min tree$^{-1}$, compared to 2.5 min tree$^{-1}$ in Emilio Baró st., or 3.0 min tree$^{-1}$ in Naranjos Av.

### 3.3.4. SMTA

This equipment has some advantages over the other two: it that can move along narrow streets, even work with parked cars if protective canvas is used, it is also cheaper than tractor mounted equipment and consequently a reduced team of workers can manage it efficiently. Like the trunk shaker, the harvesting process consists of two independent operations, a) fruit detaching, and b) fruit collecting, and transferral to the transport unit.

One or two shakes were applied to each tree, the average duration of these vibrations was 47 s (±15 s). The manoeuvres to unfasten the trunk took 11 s (±5 s), the displacement from one tree to the next 46 s (±46 s) and the approximation and fastening of the next trunk 48 s (±25 s), in total 2.5 min tree$^{-1}$ (±0.8 min tree$^{-1}$).

A team of six people carried out all the operations of traffic control, picking up the fruits from the ground, and transporting them to a container placed at 50 m. So, a team of six workers was able to collect the fallen fruits at the same rate as the shaker detached them, which is a harvesting rate of 4 tree h$^{-1}$ by a single worker.

### 3.3.5. Field Capacity Comparison

Comparing all the systems tested, it is possible to observe that the manual system is the least efficient, with only 0.63 tree $h^{-1}$ and operator (Table 3).

**Table 3.** Harvesting costs with each equipment.

| | Detaching | | | Picking | | | Total | |
|---|---|---|---|---|---|---|---|---|
| **System** | **€ $h^{-1}$** | **Tree $h^{-1}$** | **€ Tree$^{-1}$** | **Num. Workers** | **Tree $h^{-1}$** | **€ Tree$^{-1}$** | **Tree $h^{-1}$** | **€ Tree$^{-1}$** |
| Manual | 10 | 0.75 | 13.3 | 1 | 3.75 | 2.7 | 0.63 | 16.0 |
| Trunk shaker | 65 | 50.7 | 1.3 | 14 | 50.7 | 2.8 | 50.7 | 4.1 |
| T. shaker + umbrella | 70 | 20.2 | 3.5 | 7 | 20.2 | 3.5 | 20.2 | 7.0 |
| SMTA | 45 | 24 | 1.9 | 6 | 24 | 2.5 | 24 | 4.4 |

The trunk shaker is the fastest equipment (50.7 tree $h^{-1}$), but to work at this rhythm a big team of pickers (14 workers) is necessary to collect all the fruits detached.

The SMTA, can harvest over 24 tree $h^{-1}$ and requires a relatively low number of simultaneous workers collecting the fallen fruits (six workers).

The trunk shaker, being mounted on a tractor, will have problems to shaking some trees if the approximation space is narrow, meanwhile the SMTA, as can be mounted on a pedestrian tractor, and so is able to shake almost all the trees.

The shaker with umbrella has the advantage of catching most of the detached fruit, and, its reservoir was also used by the workers as an intermediate loader, as they can unload the buckets into the umbrella at ground level, and the tractor elevates the load to the lorry box. The most significant limitation of this equipment is its big size, it can only move along the street, and a lot of trees cannot be harvested due to their proximity to street furniture.

### 3.4. Tree Damages

No damages to the trees were noticed, no de-skinning, minimum defoliation, and no branch or trunk breakages. De-skinning happened occasionally in the trials done late, in May, a problem well known from agricultural citrus experiences, because this is when tree sap is moving and the skin is very sensitive [7] (Figure 13). Trees have been harvested for three years (2017, 2018, and 2019) and no problems have been detected.

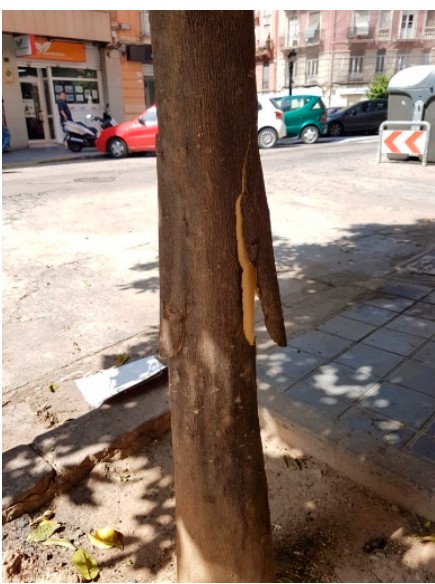

**Figure 13.** De-skinning in late harvesting (May, 2017).

### 3.5. Economical Study

Equipment rental prices in 2017, including machine and tractor driver, were 70 € h$^{-1}$ for the tractor with umbrella shaker; 65 € h$^{-1}$ for the tractor with trunk shaker; and 45 € h$^{-1}$ for the pedestrian tractor with the experimental shaker plus a driver.

To work safely, the trunk shaker and SMTA needs two auxiliary operators to control the traffic of vehicles and pedestrians in the working zone. In the case of the shaker with umbrella, an additional operator is necessary to monitor the folding and unfolding of the umbrella near houses.

With respect to picking up the fruits from the ground, in the case of the shaker with umbrella, a team of four people was necessary to collect all the oranges that fell outside the umbrella, so, this equipment uses a team of seven workers.

In the case of the trunk shaker, a team of twelve workers will be necessary to collect the oranges from the ground at the same speed as the shaker advance, plus two people controlling the traffic, 14 workers in total.

Considering the cost of labour at 10 € h$^{-1}$, the manual system is the most expensive with 16.0 € tree$^{-1}$ (Table 2), the shaker with umbrella has a cost of 7.0 € tree$^{-1}$, and finally the trunk shaker and the SMTA has an estimated cost of 4.1 and 4.4 € tree$^{-1}$ respectively.

## 4. Discussion

### 4.1. Trunk Shaker

The trunk shaker was mounted on an orchard tractor, being able to move along pedestrian zones and to avoid obstacles. Its working capacity was 1.2 min tree$^{-1}$, although a more realistic figure would be near 2 min tree$^{-1}$. An elevated number of operators must work simultaneously with the shaker to pick up the fruits from the ground. The detachment percentages and working capacity have been similar to the obtained with eccentric mass shakers in agricultural citrus crops [2–4,7,8].

### 4.2. Trunk Shaker with Umbrella

This system was the most suitable for long, wide avenues, where traffic control is easier, furthermore, it can catch most of the detached fruit and also serve as an auxiliary loader for the picking workers. On the other hand, a lot of trees cannot be harvested due to the presence of obstacles. Working capacity oscillates between 2 to 5 min tree$^{-1}$, with an average of 3 min tree$^{-1}$.

### 4.3. SMTA

This system is the slowest at detaching, with 0.5 min tree$^{-1}$, and the displacements between trees are slow if the distances are far, in total it needed 2.5 min tree$^{-1}$, but its main advantage is that can move around almost all the spaces and, therefore, almost all the trees in the city can be detached mechanically. The detachment percentages and the working capacity obtained in ornamental citrus trees, with this equipment, are better than the registered in agricultural citrus trees. This fact is due to the long trunks of the ornamental trees allow to shake the trunk, meanwhile the agricultural Mediterranean citrus trees must be shaken by the branches.

## 5. Conclusions

All the equipment reached detaching percentages over 80% and did not produce damages to the trees when they were used prior to the sap movement at the end of April.

All the mechanical systems allow a high reduction of harvesting costs in comparison with manual harvesting.

Nowadays, urban citrus trees in Valencia are being harvested with trunk shaker plus umbrella, which demonstrates the utility of this technique.

However, an important proportion of trees cannot be harvested by this system, because the large dimensions of the equipment impede it. The construction of smaller sized equipment, like SMTA, that can be mounted on pedestrian or small tractors, is therefore of great interest.

**Author Contributions:** A.T. conceived and designed the conceptual framework of the study; A.T., E.O., M.P., and J.M.M. performed the field experiment, designed and constructed a prototype and acquired data; P.X. managed the biological aspects; C.O. and A.T. analyzed and interpreted the results and wrote the original paper. All authors revised and approved the final manuscript

**Funding:** This research was funded by research project RTA2014-00025-C05-02 financed by Instituto Nacional de Investigaciones Agrarias (INIA) and European FEDER founds.

**Acknowledgments:** Our acknowledgements to Ayuntamiento de Valencia and the technicians: Juan José Peña, Diego Guerra, Pablo Valverde, Andrés Cuenca and Bryan Pacheco.

**Conflicts of Interest:** The authors declare no conflict of interest. The funders had no role in the design of the study; in the collection, analyses, or interpretation of data; in the writing of the manuscript, or in the decision to publish the results.

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
