# Peer review of "Mechanical Harvesting of Ornamental Citrus Trees in Valencia, Spain"

_agronomy, doi:10.3390/agronomy9120827_

Round 1
Reviewer 1 Report
This paper describes a comparison study for evaluation of 3 citrus trees harvesting mechanisms versus manual detachment. The experiments provide an analysis for a variety not often being analyzed in harvesting literature due to its ornamental nature (vs trees grown in orchards for food production), and provide financial and qualitative results. The results are interesting, the experiments are clearly described an reproducible. The paper written in a clear manner, with technical details provided.
Given my positive attitude I was missing structuring of the writing of the paper as a scientific paper and some details were missing as outlined bellow. Bellow are my suggestions.
Referencing – While it is clear the decisions made for mechanical harvesting are based on current literature, the paper is missing discussion of other researchers ongoing work. The review paper by Sanders et al., (5) for example is briefly cited with mentioning very little about, while most of the method is based, it seems on common practices mentioned in their review. Vibrational parameters are set based on the works by Torregrosa, Whitney and Ortiz but very little is mentioned on what this means and how these methods compare. I believe the paper will benefit from inclusion of this information for a clearer overall image of the state of the art (no need to turn this into a review paper of course, but some discussion to give the results a background would be beneficial).
Discussion of comparison of the results obtained to the state of the art is missing in the discussion section, especially how these results are compared to the similar methods applied to non ornamental varieties.
Figures 1-4 – While the image gives a good overview of the setting it is hard to obtain an informative description of the actual shaker. Maybe a more close up photo of the tools or some sort of a sketch will be beneficial to understand how this system works (to accompany the textual description)
Line 65 - “1 to 3 shakes were applied, depending on the amount of fruit remaining on the tree...” - Please clarify the decision process, when it is decided to take another shake. Seems like a critical point to understand the results.
Line 111-112 – Please clarify how the mass was measured and what is 50 N.
Sections 2.2-2.3 provide a good overview of the setting, there is though lack of description of the final dataset such as sample size (how many trees were examined in the end, how many fruits) it is also not clear if this experiment was done once for each area measured or was their repeatability (some results are shown over 2 years 2018-2019). Some of the information appears in the results section, but in a scattered manner making it hard to keep track of all the details.
Data – have you considered releasing the acquired data? Especially from the camera and the ground truth of number of fruits on a tree before and after shaking
In my opinion section 3.1 is not part of results but of methods (though it is the measured shaking, it is not what in my opinion the paper is about, but the detachment results and the financial analysis)
Great table 1 – really helps the readability of this section (maybe consider having a similar table for sections 2.2-2.3? with the additional information mentioned above). Also, maybe reference to it earlier on will help readability, and maybe shortening the number of repeated details in the text and in the table (such as frequency), by referring to the table.
Figure 9 – x axis numbers are unreadable, please consider moving the axis labels or make other changes in the design.
Section 3.2.3 and figure 10, what fitting curve was used? Is this a logarithmic fitting?
Section 3.3.1 describes analysis of the video and gives out final numbers of picking and detaching rates. Not clear what methodology was used to obtain this information, how many videos were analyzed, who was looking at these videos, etc… please provide more information. Additionally the numbers reported are in fr/min while the rates in equations 2+3 are in min/fr, suggest using the same in both (e.g. 5 fr/min = 0.2 min/fr), Also the rounding done in equation 3 is inaccurate – 23 fr/mn is 0.0435 min/fr which leads to 3.45 trees/h and not 3.75…
Overall English level is fine. But some English/writing style corrections are needed to keep the language publishable such as:
Longer paragraphs (that don’t consist of a single sentence at a time) might help the reading flow
Past and current tenses are mixed
Excessive usage of “etc”
Line 29: “*based* on personal communications with….”
Line 30: Done by hand = performed manually.
Line 81: “This size of *this * equipment”.
Line 268 has an extra parentheses
Suggest the paper goes through language editing.
Author Response
"Please see the attachment."

Reviewer 2 Report
In this paper the effect of different mechanical Harvesting methods have been evaluated. The paper is scientifically interesting but some improvement could be done:
Introduction can be improved considering the importance of adopting mechanical harvesting technique also in other fields, as well as recent research in the characterization of olive.
Miglietta, R. Micale, R. Sciortino, T. Caruso, A. Giallanza, The sustainability of olive orchard planting management for different harvesting techniques: An integrated methodology, Journal of Cleaner production, 2019, vol 238
Marchese A., Marra F.P., Caruso T., Mhelembe K., Costa F., Fretto S., Sargent D.J., The first high-density sequence characterized SNP-based linkage map of olive (Olea europaea L. subsp. europaea) developed using genotyping by sequencing, (2016) Australian Journal of Crop Science, 10 (6) , pp. 857-863.
Di Vaio C., Marra F.P., Scaglione G., La Mantia M., Caruso T.,The effect of different vigour olive clones on growth, dry matter partitioning and gas exchange under water deficit (2012) Scientia Horticulturae, 134 , pp. 72-78.
Discussion needs to be improved. I think that a table ables to synthesize all the results could be useful in highlighting the benefits related on the use of mechanical tools.
